# Social disparities in unplanned 30-day readmission rates after hospital discharge in patients with chronic health conditions: A retrospective cohort study using patient level hospital administrative data linked to the population census in Switzerland

**Andrea Zumbrunn**[1]*, **Nicole Bachmann**[1], **Lucy Bayer-Oglesby**[1], **Reto Joerg**[2], on behalf of the SIHOS Team[¶]

**1** Institute for Social Work and Health, School of Social Work, University of Applied Sciences and Arts Northwestern Switzerland, Olten, Switzerland, **2** Swiss Health Observatory, Neuchâtel, Switzerland

¶ Membership of the SIHOS Team is provided in the Acknowledgments
* andrea.zumbrunn@fhnw.ch

## Abstract

Unplanned readmissions shortly after discharge from hospital are common in chronic diseases. The risk of readmission has been shown to be related both to hospital care, e.g., medical complications, and to patients' resources and abilities to manage the chronic disease at home and to make appropriate use of outpatient medical care. Despite a growing body of evidence on social determinants of health and health behaviour, little is known about the impact of social and contextual factors on readmission rates. The objective of this study was to analyse possible effects of educational, financial and social resources of patients with different chronic health conditions on unplanned 30 day-readmission risks. The study made use of nationwide inpatient hospital data that was linked with Swiss census data. The sample included n = 62,109 patients aged 25 and older, hospitalized between 2012 and 2016 for one of 12 selected chronic conditions. Multivariate logistic regressions analysis was performed. Our results point to a significant association between social factors and readmission rates for patients with chronic conditions. Patients with upper secondary education (OR = 1.26, 95% CI: 1.11, 1.44) and compulsory education (OR = 1.51, 95% CI: 1.31, 1.74) had higher readmission rates than those with tertiary education when taking into account demographic, social and health status factors. Having private or semi-private hospital insurance was associated with a lower risk for 30-day readmission compared to patients with mandatory insurance (OR = 0.81, 95% CI: 0.73, 0.90). We did not find a general effect of social resources, measured by living with others in a household, on readmission rates. The risk of readmission for patients with chronic conditions was also strongly predicted by type of chronic condition and by factors related to health status, such as previous hospitalizations before the index hospitalization (+77%), number of comorbidities (+15% higher probability per additional comorbidity) as well as particularly long hospitalizations (+64%).

**Data Availability Statement:** Anonymized individual data from different data sets were used for the construction of the SIHOS database. All these data are the property of the Swiss Federal Statistical Office (SFSO) and can only be made available by legal agreements with the SFSO. The data that support the findings of the present analysis are used under license for the SIHOS study and are not publicly available due to the data protection restrictions. However, they are available after signing a data protection contract with the Swiss Federal Statistical Office (SFSO) Sektion Gesundheitsversorgung, Espace de l'Europe10, CH-2010 Neuchâtel, Switzerland Phone:+41 58 463 67 00 Email: gesundheit@bfs.admin.ch. For a detailed description of the different data sources and the linkage procedure see Bayer-Oglesby et al. (2021).

**Funding:** This work was supported by the SNSF National Research Programme "Smarter Health Care"(NRP74), project number 4, grant number 407440_167506, applicant Lucy Bayer-Oglesby. Project and funding description are available at http://www.nfp74.ch/en/projects/in-patient-care/project-bayer-oglesby. The funder had no role in the study design, data collection and analysis, decision to publish, or preparation of the manuscript. The views reported here are the authors' views and do not necessarily reflect the funding organization.

**Competing interests:** The authors declare that they have no competing interests.

Stratified analysis by type of chronic condition revealed differential effects of social factors on readmissions risks. Compulsory education was most strongly associated with higher odds for readmission among patients with lung cancer (+142%), congestive heart failure (+63%) and back problems (+53%). We assume that low socioeconomic status among patients with chronic conditions increases the risk of unplanned 30-day readmission after hospitalisation due to factors related to their social situation (e.g., low health literacy, material deprivation, high social burden), which may negatively affect cooperation with care providers and adherence to recommended therapies as well as hamper active participation in the medical process and the development of a shared understanding of the disease and its cure. Higher levels of comorbidity in socially disadvantaged patients can also make appropriate self-management and use of outpatient care more difficult. Our findings suggest a need for increased preventive measures for disadvantaged populations groups to promote early detection of diseases and to remove financial or knowledge-based barriers to medical care. Socially disadvantaged patients should also be strengthened more in their individual and social resources for coping with illness.

## Introduction

Unplanned readmissions shortly after acute hospital discharge are generally seen as unwanted events as they represent an instable and deteriorating health status of patients. They occur frequently in patients with chronic health conditions (CHC), such as chronic obstructive pulmonary disease (COPD) or congestive heart failure (CHF) [1]. Chronic health conditions are the focus of this article since they make up 80 percent of total healthcare costs in Switzerland [2]). Unplanned readmission rates have been used in many countries as quality indicator for hospital care, also in Switzerland [3], since they may be the result of suboptimal clinical processes provided by the hospital, and, as such, preventable. This includes failures at the level of integrated care, such as coordination and communication problems or inappropriate discharge settings, failures related to diagnostics, medication or surgical procedures or due to shortcomings in the skills or knowledge of the care provider [4]. Unplanned readmissions are also due to factors beyond the scope of hospitals. For instance, effective outpatient care has been shown to reduce the likelihood of readmission after hospital discharge [5]. This accounts in particular for health conditions known as ambulatory care-sensitive (ACS) conditions for which hospital care is expected to be preventable by adequate use of primary health care and successful self-management [6,7]. Coping successfully with a chronic health condition includes adherence to medical instructions, such as following a prescribed therapeutic regimen and taking prescribed medication [7,8]. It also requires knowledge and skills of patients, such as adequate and varying ways of coping with symptoms, distress and the fear of consequences [9] as well as the capacity to make optimal choices and participate actively and responsibly in the medical care process [10].

However, self-management of chronic illness is too often discussed from a purely individualistic perspective, ignoring the social and cultural context in which this process takes place [11,12]. Several studies have found that a person's social context not only contributes to the course and outcome of an illness, but also to their ability to cope with it [7,13]. In particular, people's socioeconomic position in society has been shown to have a strong influence on the conditions of their daily lives, leading to an unequal distribution of the resources and skills

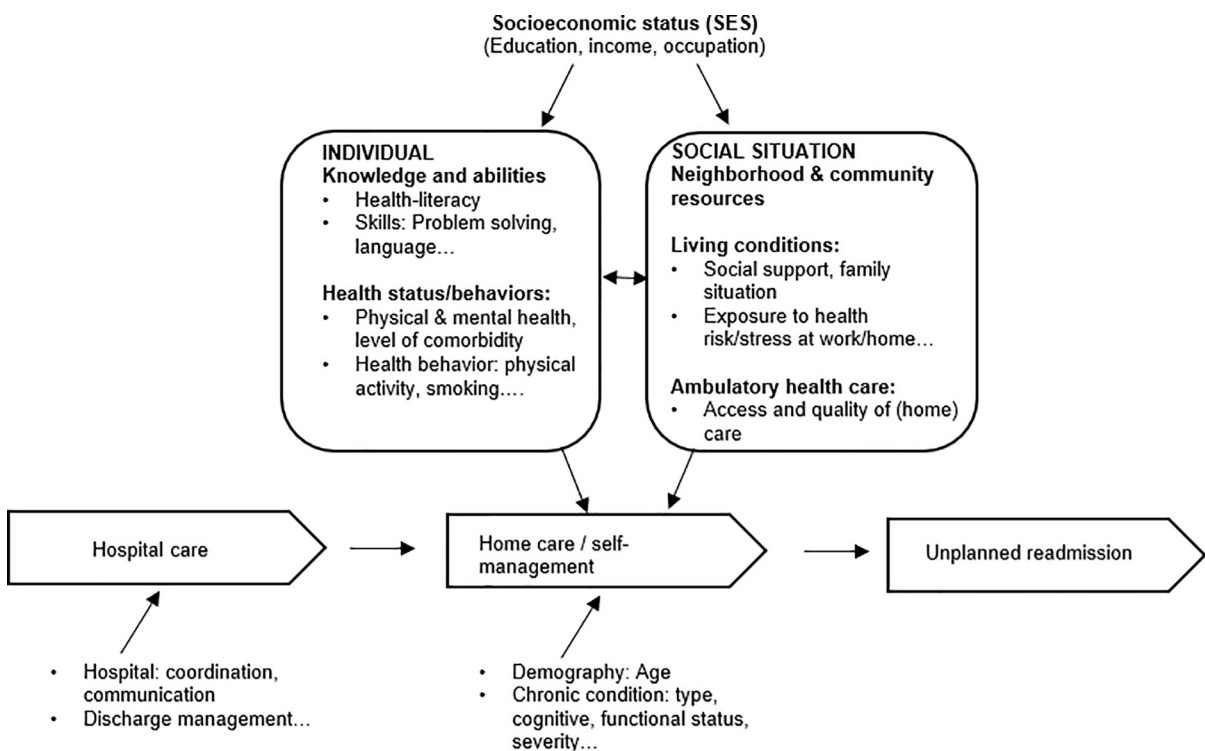

**Fig 1. Conceptual framework on social determinants of health influencing self-management and hospital readmissions [13,20,27].**

needed to cope with the illness [14]. Fig 1 summarizes key social determinants of health that may influence self-management and hospital readmission. Firstly, socially disadvantaged groups are known to have lower overall health literacy [15–17], resulting presumably in limited understanding of disease and its relationship to one's behaviour, as well as a lower ability to adhere to recommended lifestyle changes. Secondly, socially disadvantaged groups are less likely to be in good health [18,19] and show more frequent unhealthy behaviours such as smoking and inactivity [20], which might additionally complicate coping with chronic conditions. Thirdly, socially disadvantaged groups are also more often exposed to stressful living conditions, such as high demands and low individual control over work [21] or stressful family relationships or family obligations that are considered more important than the chronic condition [13,20]. Social support from family members and friends has also been shown to facilitate coping with a chronic disease [10,22], but socially disadvantaged groups are less likely to have this support [23,24]. Finally, socially disadvantaged groups have less access to adequate preventive and health care services [19]. Despite a universal health insurance system that is compulsory and that covers ambulatory and hospital care, some services, such as home care services after hospital discharge, are not covered in Switzerland, which makes financial resources a precondition for access to professional home care. Poor communication between different health care providers and patients/proxies [25] or the lower likelihood of seeing a primary care provider or the appropriate specialist after discharge from hospital [26] have also been observed, especially for patients from ethnic minorities or/and with limited language skills.

Despite this knowledge, the effects of social factors on unplanned readmission rates of the chronically ill have been studied insufficiently and study results are inconsistent. In a systematic review on readmissions after hospitalization for pneumonia and heart failure, *educational level* predicted readmission rates only in one study (from n = 3) in multivariate analyses [28].

A recent study with COPD-patients, accounting for multiple social and clinical factors, also found no effect of educational level on 30-day readmission rates [29]. More pronounced evidence is available for indicators of a person's *financial situation*. In the above-mentioned review, insurance type, income or socioeconomic status (SES) were found to be independent predictors of the 30-day readmission rate in 6 (out of n = 9) studies [28]. Household income and insurance status also significantly influenced 30-day readmission in patients with early stage lung cancer [30]. For COPD-patients, both positive associations [31] and no associations were found [29]. Few studies take a socio-ecological perspective on readmission rates by including indicators of social disparities at the neighbourhood or community level. For instance, in a study by Foraker et al. [32], heart failure patients living in low median income areas were readmitted at a higher rate than patients from high median income areas when controlling for educational degree and comorbidity. On the other hand, in an analysis of population-based administrative data of non-psychiatric hospitalizations, average neighbourhood income was no longer associated with 30-day urgent readmission after accounting for known health and social risk factors using a validated index (LACE+) [33]. Other publications discuss effects of socioeconomic factors in the context of quality measures and reimbursement systems of hospitals but do not focus exclusively on chronic conditions [34–37]. According to Obuobi (2021) [36], who analysed all patients at an urban tertiary care centre, socially disadvantaged patients appear to be more severely sick, but the higher level of illness is adequately reflected in the reimbursement system. Other studies found direct effects of social factors on readmission rates in addition to higher comorbidity and thus advocate adjusting the reimbursement system for social risk factors [37]. The higher burden of disease is consistent with the findings of several studies on the higher likelihood of multimorbidity and comorbidity among groups with low educational attainment [38,39] and low income [40] and is in line with the research on social health determinants mentioned earlier.

Regarding *support of social networks*, there is some evidence that support of chronically ill patients by informal caregivers decreases the risk of unplanned readmissions. In the above mentioned systematic review on readmissions after hospitalization for pneumonia and heart failure, lack of social support and being unmarried were significantly associated in multivariate analysis with a higher readmission risk in three out of five studies [28]. In a clinical sample, patients living alone were over three times more likely to be readmitted within 30 days to hospital after coronary artery bypass graft surgery than those living not alone [41]. A higher risk for 30-day readmission in CHF patients was also found for unmarried patients, who are more likely to live alone [42]. On the other hand, in a study in two urban hospitals, which included also non-chronically ill patients, living alone showed no effect on the potentially preventable hospital readmission rate [43]. Neither showed living alone and care giver availability as having an effect on the 30-day readmission rate in severe COPD patients when controlling for health status and further social factors [29].

In summary, the current literature on social factors affecting the post-discharge period and in particular readmission rates of chronically ill patients is inconsistent and an overall picture is still missing. This is partially due to limitations of study designs. Nationally representative samples are scarce since hospital register data rarely contain information on the social situation and living arrangement of discharged patients. Several study results are restricted to one particular chronic health condition, do not focus on chronic conditions or have little explanatory power due to small sample sizes or single-centre study designs. Since social disparities are often not the main study objective in clinical trials and do not determine study design, (missing) effects of social factors are therefore difficult to interpret. Possibly, the inconsistent findings may also be the result of different assessment methods of social indicators, especially the

use of neighbourhood measures as a proxy for individual SES as aggregated data may not reflect effects of social factors as accurately as individual level data.

The present study aims to analyse differences in readmission rates between social groups and in doing so, to gain a better insight into the complexity of unplanned readmissions processes by separating the impact of health status from social factors. To overcome the limitations of previous studies, analysis is performed on the individual level based on Swiss hospital administrative data, which is linked to a national representative sample of population census data. The analysis comprises various non-communicable diseases (NCD) which allows comparisons between chronic conditions. Based on current research, we hypothesize that patients with low resources (education, financial, social) are at higher risk for unplanned 30-day readmission after hospital discharge than patients with high resources. Pronounced social differences in readmission rates are to be expected, especially for chronic conditions that are sensitive to ambulatory care and rely on a high level of personal and social resources for self-management (ACS conditions).

The paper addresses the following questions:

1. Are the social characteristics of chronically ill inpatients, such as educational level, financial and social resources, significantly correlated with the risk of an unplanned 30-day readmission in Switzerland if demographic factors, health status (chronic condition, level of comorbidity and previous hospitalizations) and length of stay in hospital are controlled for?

2. Does the effect of social characteristics of chronically ill patients on the risk of an unplanned 30-day readmission differ between ACS vs. non-ACS-conditions if demographic factors, health status (level of comorbidity and previous hospitalizations) and length of stay in hospital are controlled for?

## Methods

We followed recommendations for the reporting of studies using observational routinely-collected health data (RECORD) [44] and guidelines for regression analysis in medical research [45].

### Data sources and study population

The database was created within the research project "Social Inequalities and Hospitalizations (SIHOS)" which is part of the National Research Program 74 "Smarter Health Care" (www. nrp74.ch). For the SIHOS database census data from the Structural Survey (SE) was linked to a dataset from the medical statistics of hospitals (MS) for the first time in Switzerland, resulting in a unique retrospective cohort study. The SE is a representative sample of around 200,000 persons each year living in private households. In the MS all hospital discharges in Swiss hospitals are registered on a mandatory basis. The MS contain medical, sociodemographic as well as administrative information on the individual patient level. By using the anonymous linkage code, 1.2 million records from SE (years 2010–2014) were combined with 9.6 million records from the MS (years 2010–2016) resulting in the SIHOS database containing 950,182 successfully linked records of individuals 15 years and older living in Switzerland in a private household and who had been hospitalized in an acute hospital at least once between 2010 and 2016.

Extensive validation revealed that hospitalization rates observed in the SIHOS database are around 35% lower than the actual rates of the Swiss population because of (1) incorrectly generated anonymous linkage codes, (2) under-representation of individuals with severe health problems in the SE sample and (3) demographic deviations of the SE sample from the reference population. However, since the underrepresentation concerned all social groups equally,

the successful linking of hospitalizations to individuals can be regarded as a random selection of the study population with the exception of some non-European migrant groups (0.7% of entire sample, 5.9% of non-Swiss sample). For the latter the linkage failed disproportionately often due to inconsistent writing of unfamiliar names. Thanks to the linkage of MS and SE, the SIHOS database includes information on the social situation, health status and hospital stays of a nearly representative sample of the source population. The database is used under license and is not publicly available due to data protection restrictions unless a data protection contract is signed with the Federal Office of Statistics (FOS) (see Bayer-Oglesby et al., 2021 [46] for details).

*Study sample.* Since the definition of readmission changed in the MS in 2012 [47], records of the MS 2010 and 2011 were excluded to guarantee comparability (Fig 2). Patients younger than 25 years of age were excluded since education level is a meaningful value only from around the age of 25 upwards [48]. As coping with chronic conditions at home shortly after hospital discharge was the research interest, we excluded patients hospitalized in rehabilitation wards, referrals or discharges to acute-care hospitals or long-term care institutions, clinics and residential homes. The chronic diseases include cancers, diabetes, cardiovascular, respiratory and musculoskeletal diseases and are among the leading chronic conditions in high-income countries according to disability-adjusted life years [49]. They were selected in accordance with the following criteria: (1) Chronic condition or an acute incident of a chronic condition, (2) frequency of the disease in Switzerland, (3) frequency of hospitalizations due to the disease in Switzerland, and (4) percentage of all deaths caused by the disease in Switzerland. We included back problems and osteoarthritis in the comparison to take into account different risk factors [50]. The 12 selected chronic conditions were defined according to ICD10-GM codes and Clinical Classification Software (CCS Level 1) (cf. Table 1). The index hospitalization was defined as the first inpatient hospital stay for one of the selected chronic conditions during the study period. Therefore, a patient was counted only once during the observation period to avoid overrepresentation of social factors of patient groups with frequent discharges. Patients were excluded if the observation period after hospital discharge was less than 30 days (or unknown or implausible) to determine whether or not readmission occurred within 30 days. Patients who had died during hospital stay or within 30 days after discharge were excluded. The sample therefore consisted of individuals who were at least 25 years old at the time of hospitalization and who had at least one hospital discharge to their own home for a somatic chronic health condition between 2012 and 2016.

The Northwestern and Central Switzerland Ethics Committee confirmed that the SIHOS study is exempted from ethics committee approval according to the Swiss Human Research Act, because it is based on anonymized administrative data (2017–01125).

## Variables

We defined the *outcome variable* as unplanned readmission within 30 days after hospital discharge using information on the type of admission ("Emergency admission, treatment within 12 hours indispensable": 1: yes, 0: no), which is a common practice in Switzerland [47,51]. Planned readmissions were not looked at since they may be part of the treatment. We looked at all-causes readmissions since comorbidity is common in chronic conditions and readmissions are often caused by diseases other than the index hospitalization [1]. Moreover, additional conditions may have not been detected at index hospitalization. Preventability of readmission from a medical perspective, e.g., narrowing it down to complications of the initial disease, was therefore not considered. Readmissions to the same or to another hospital were both taken into account. Misclassifications of the outcome variable are possible if the matching

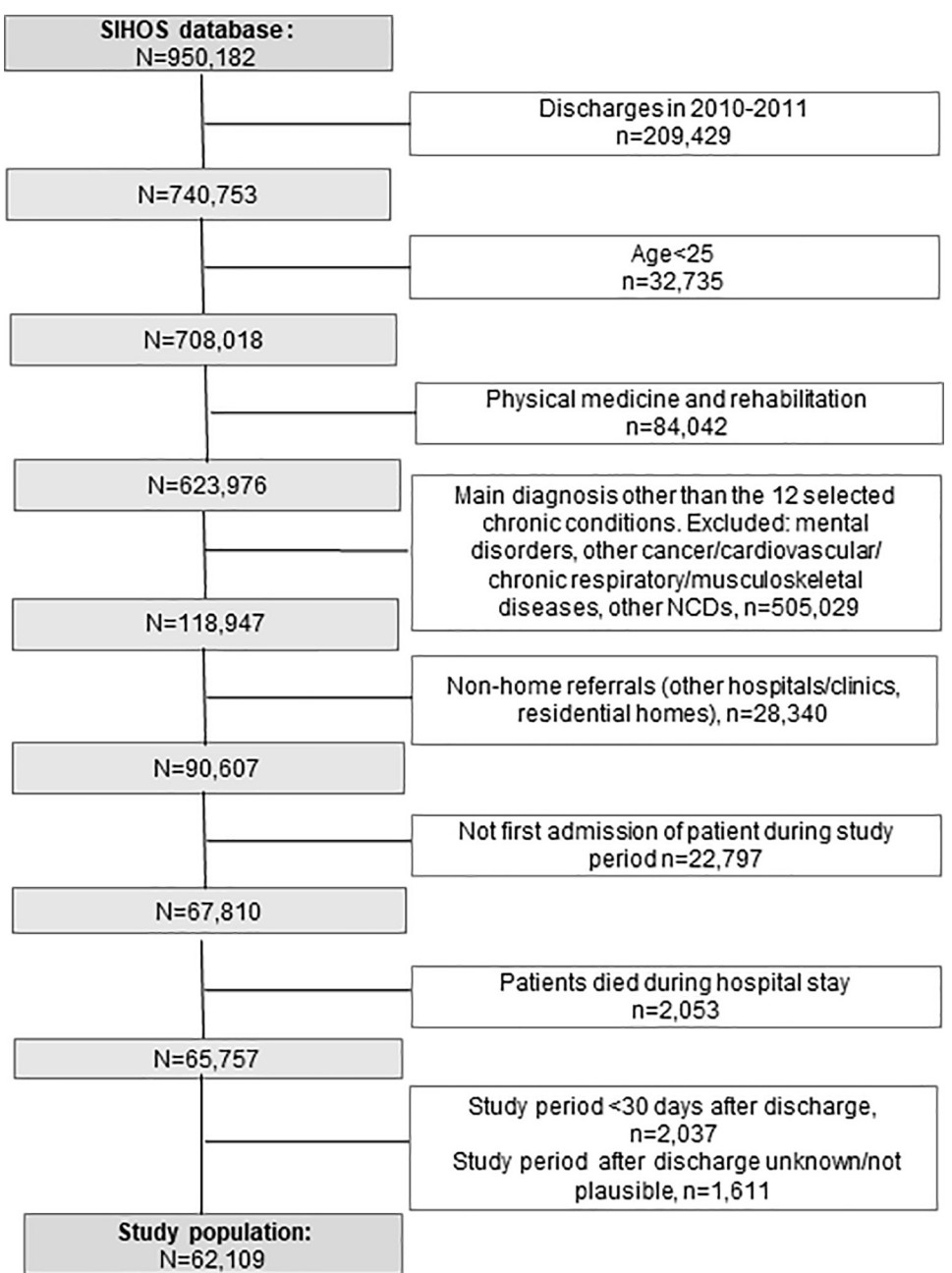

**Fig 2. Flow chart of the selection process for the study sample.**

of discharges to readmissions is incorrect. Because of extensive validation of the matching process and the exclusion of implausible cases, the risk of such misclassification bias is very small [46].

*Health status of patients* is captured by five variables. In the all-conditions model (cf. statistical analysis), we included the main diagnosis of the index hospital stay for one of the 12 selected chronic diseases, cf. Table 1 (1: yes, 0: no). Comorbidity was controlled for to reflect the patient's co-occurring conditions not directly tied to the index hospitalization's principal diagnosis (5). Based on the works of Elixhauser et al. [52], comorbidities were defined as side

**Table 1. Definition of chronic health conditions.**

| Specific chronic diseases (main diagnosis) | CCS Level 1* | ICD10-GM Codes (Vers. 2017) |
|---|---|---|
| Malignant neoplasms (cancer) | | |
| Lung cancer | CCS_LEV1 = 19 | C34, D022 |
| Colon cancer | CCS_LEV1 = 14 | C18, D010 |
| Breast cancer (women) | CCS_LEV1 = 24 | C50, D05 |
| Prostate cancer (men) | CCS_LEV1 = 29 | C61, D075 |
| Cardiovascular diseases (incl. risk factors) | | |
| Diabetes | CCS_LEV1 = 49<br>CCS_LEV1 = 50 | E109, E119, E139, E149, R73, excl. E12 (Diabetes related to Malnutrition); E10-E14; 3rd/4th decimal place for complications (excl. 3rd decimal place = 9 = w/o complication) |
| Congestive heart failure (CHF) | CCS_LEV1 = 108 | I50 |
| Ischaemic heart disease | CCS_LEV1 = 101 | I20, I24, I25 |
| Acute myocardial infarction (AMI) | CCS_LEV1 = 100 | I21, I22 |
| Acute cerebrovascular diseases | CCS_LEV1 = 109 | I60-I64, I66 |
| Chronic respiratory diseases | | |
| Chronic obstructive pulmonary disease (COPD), incl. asthma | CCS_LEV1 = 127<br>CCS_LEV1 = 128 | J40-J44, J47<br>J45, J46 |
| Musculoskeletal diseases | | |
| Osteoarthritis | CCS_LEV1 = 203 | M15-M19 |
| Back problems | CCS_LEV1 = 205 | M43.2, M43.3, M43.4, M43.5, M43.6, M45, M46 (excl. M46.2, M46.3), M47, M48 (excl. M48.5), M49 (excl. M49.0, M49.5), M50, M51, M53, M54 |

*CCS = Clinical Classifications Software; developed by the Healthcare Cost and Utilization Project (HCUP), financed by the US-Agency for Healthcare Research and Quality, adapted for Switzerland by Daniel Zahnd, Bern University of Applied Sciences of Applied Sciences.

diagnoses known to influence outcome indicators in hospitals admissions such as mortality and readmission rates. In accordance with Moore et al. [53] and in contrast to the original work [52], cardiac arrhythmia was not considered, uncomplicated and complicated hypertension/diabetes were combined and psychiatric diagnoses excluded. Based on the remaining comorbidity diagnoses, the number of somatic comorbidities (NSC) was used as measure of comorbidity, either as a metric measure centred by the main diagnosis allowing us to control for different means and within-group variability (all-conditions model) or categorized in (0) none, (1) one, (2) two, (3) three and more somatic comorbidities (condition-specific models, cf. statistical analysis). Mental comorbidity (1: yes, 0: no) was introduced separately in the model. Hospital stays in the previous 6 months for any condition other than the chronic condition (1: yes, 0: no) controlled for unstable health conditions. Length of stay in hospital (LOS) was calculated based on the SwissDRG definition by day of admission and each subsequent day without the day of discharge and excluding days of leave [54]. LOS was centred by the main diagnosis, allowing us to control for different means and within-group variability, and categorized in quantiles (all-conditions model) or dichotomized (4th quantile, 1: yes, 0: no) (disease-specific models). Finally, the variable "year of hospital discharge" was available to account for variation in overall readmission rates during the study period due to financial and administrative processes in hospitals [55].

*Social characteristics* of patients were analysed by three predictor variables. Firstly, formal education level as a classic indicator of vertical social inequality was categorized into tertiary level qualification for university and college degrees (0), upper secondary education degree, mainly vocational education (1) and no further qualification than nine or less years of compulsory school (2). This is a meaningful value from around the age of 25 upwards [48]. Secondly, household type was operationalized as living with others (0) and living alone (1) to indicate

options for social support in the case of health problems provided by people living in the same household. Living alone has been discussed as a risk factor for receiving little social support and for feelings of loneliness [23], although living alone does not preclude a person from having a large, strong social network. Thirdly, hospital insurance class, general (0) vs. semi-private and private (1), was used as crude proxy for income, since (semi-) private insurance status correlates with income in the Swiss population [56]. It may also have an impact on the type and volume of medical interventions during the hospital stay [56,57]. Demographic patient characteristics were included as covariates, i.e., female sex (1) and age groups (25–54, 55–65, 67–74, 75+) in the all-conditions model and in years in condition-specific models. Not speaking (1) vs. speaking the regional language (or English) (0) was included in the analysis as one possible hindering factor in the hospital context for understanding and following recommendations after discharge. Nationality was excluded because of collinearity with language skills.

## Statistical analysis

Statistical analysis was performed using IBM SPSS Statistics Version 27 and Mlwin Version 3.01. Univariate comparisons of independent variables between patients with and without 30-day readmission were carried out using $Chi^2$-tests in case of categorical variables and two-sample t-tests in case of continuous variables. Frequency and graphical procedures were used to document descriptive characteristics of the study sample. Associations between independent social factors and the likelihood of 30-day readmission, adjusted for covariates, were analysed with multivariate logistic regression models. Variables whose bivariate association with the outcome had a p-value > 0.25 were excluded from multivariate models [58]. Year of discharge was the only such variable ($Chi^2$, 3.13(4), p = 0.537). Predictors and covariates were entered in three steps: Model A includes social factors educational attainment, insurance class and household type and controls for demography (sex, age, language skills). Model B introduces health status factors, namely the main diagnosis (one of the selected 12 chronic diseases) in all-conditions model, the number of Elixhauser comorbidities, mental comorbidity and previous hospital stays. Model C also controls for length of hospital stay. For the fully adjusted Model C, statistical interactions were tested between all social factors, between demographic and social factors and between main diagnosis and social factors by introducing the corresponding two-way interaction terms one by one. As the fit of pooled models was not satisfactory (according to the Hosmer-Lemeshow test) even when containing interaction terms, we ran separate logistic regression models for the different comorbidities, to identify possible condition-specific effects of social determinants on 30-day readmission risks. Statistical significance was set at p<0.05. Missing values were mostly due to lack of information on language skills (n = 770). In accordance with Hosmer and Lemeshow (2000) [58], we tried to prevent overfitting by constraining the number of parameters of the model to one tenth of the number of individuals in the outcome category in the stratified analyses. In order not to violate this condition, parsimonious models were calculated for condition-specific models. Analyses for chronic conditions with n<100 readmissions were omitted.

Multicollinearity between categorical variables was tested by Cramer's V. No value was greater than .3 indicating a lack of problematic associations among social variables [59]. Multicollinearity for scale variables was interpreted as absent if Pearson r was smaller than .8, VIF smaller than 5 and highest conditional index smaller than 30, which was the case for all correlations. According to the Box-Tidwell approach [60], the covariates age and length of stay did not show a linear relation with the logit of the outcome. Both variables were therefore categorized in the all-conditions model, the same accounted for the number of somatic comorbidities and length of stay in the condition-specific models. Likelihood ratio tests were used to evaluate

differences between log-likelihoods of models, and the area under the "receiver operating characteristic (ROC)" curve [61] and Hosmer-Lemeshow Test were used to compare model fit. To test the null hypothesis that a single predictor is not significant, we used the Wald test, which follows an approximately chi$^2$ distribution if the sample size is large enough.

To assess potential clustering of data, we first ran a multilevel logistic regression model to estimate the variance on the second level (hospital level, n = 178). Given the fact that the intra-class correlation in the null model was .07, dropping to .04 when social variables were added in the model, the clustering was estimated to be below the conventional threshold of .05, indicating substantial evidence of clustering [62]. This means that there are no large differences in readmission rates between hospitals. Clustering by hospitals was therefore not adjusted for. Moreover, since only index hospitalizations of patients were included, multiple hospital admissions of single patients did not present a risk for confounding on the individual level.

## Results

### Results of bivariate analysis

The patient characteristics are presented in Table 2. Of the 62,109 patients aged 25 years and older (25 to 108 years), 3.4% (n = 2,119) had an unplanned readmission within 30 days after discharge. Demographic and social characteristics of those with and without 30-day readmission differed significantly. Patients with readmission were significantly older (71.2 vs. 65.1 years), were more often men (45.8% vs. 41.7%), less often had (semi-) private hospital insurance (27.5% vs. 32.7%), lived alone more often (31.6% vs. 24.9%) and were more likely to have low educational attainment (36.3% vs. 27.5%). Significant differences were also observed regarding health status. Patients with 30-day readmission were more likely to have lung, colon or prostate cancer, diabetes, congestive heart failure or COPD as the main diagnosis than those without (see also readmission rates in Table 3). They had, on average, a higher number of somatic comorbidities (2.1 vs. 1.2), and were more likely to have a mental comorbidity (11.3% vs. 6.7%), to have been in hospital in the preceding six months (23.3% vs. 10.4%) and to have had particularly long hospital stays (4th quartile: 33.6% vs. 21.5%).

Furthermore, social groups differed in the distribution of chronic conditions (Table 3). The proportion of patients with low level of education was highest for congestive heart failure (43.3%), COPD (40.7%), diabetes (38.3%) and lung cancer (34.8%) and clearly lowest for prostate cancer (17.5%). COPD, congestive heart failure and diabetes were also the most frequent chronic conditions of patients in single households and of patients with mandatory hospital insurance when compared to their corresponding reference group. Low rates of patients with mandatory hospital insurance were observed for prostate cancer, breast cancer and ischaemic heart disease. As can be seen in Table 3, in the majority of the cases with 30-day readmission, causes of the readmissions, presented here by clinical classification group, differed from those of the index hospitalizations. In total, only 23.9% (CCS lev1) or 35.6% (CCS lev2) were readmitted to hospital for the same clinical reasons. The clinical classification of admission and readmission matched the most for the four types of cancer. Moreover, 25% of patients were readmitted to another hospital. This was most often the case for patients with ischaemic heart disease (45.5%), back problems (39.2%) and osteoarthritis (38%). The unplanned readmission took place on average 14.3 days after hospital discharge, with prostate cancer patients readmitted the earliest, on average after 9.8 days and patients with congestive heart failure the latest, after 16.2 days.

According to Fig 3, significant differences in multi/-comorbidity were observed by social characteristics. On average, socially disadvantaged patients had a higher number of somatic comorbidities defined by Elixhauser [52] than privileged patients. As can be seen in Fig 3 (top

**Table 2. Characteristics of patients with/without unplanned 30-day readmission: Demographic, social, health status and hospital stay.**

| | Total sample (n = 62,109) | no readmission within 30 days (n = 59,990) | readmission within 30 days (n = 2,119) | p |
|---|---|---|---|---|
| **SOCIAL SITUATION** | | | | |
| age (x, SD) [3] | 65.3 (13.4) | 65.1 (13.3) | 71.2 (12.7) | < .001 |
| women (%) | 45.6 | 45.8 | 41.7 | < .001 |
| no regional language skills (%) | 11.2[1] | 11.1 | 12.4 | 0.08 |
| (semi-)private (%) | 32.6[2] | 32.7 | 27.5 | < .001 |
| living alone (%) | 25.1 | 24.9 | 31.6 | < .001 |
| Educational attainment: | | | | |
| compulsory education (%) | 27.8 | 27.5 | 36.3 | |
| upper secondary/vocational education (%) | 50.8 | 50.9 | 48 | |
| tertiary education (%) | 21.4 | 21.6 | 15.7 | < .001 |
| **HEALTH STATUS** | | | | |
| **Main diagnosis** | | | | |
| Lung cancer (%) | 2.3 | 2 | 9.4 | |
| Colon cancer (%) | 2.2 | 2.2 | 3.3 | |
| Breast cancer (%) | 7.3 | 7.4 | 4.8 | |
| Prostate cancer (%) | 4.5 | 4.4 | 6.8 | |
| Diabetes (%) | 2.5 | 2.5 | 3.7 | |
| Acute myocardial infarction (%) | 6.9 | 6.8 | 7.8 | |
| Acute cerebrovascular diseases (%) | 5.0 | 4.9 | 6.7 | |
| Ischaemic heart disease (%) | 11.1 | 11.1 | 8.7 | |
| Congestive heart failure (%) | 5.6 | 5.3 | 15.3 | |
| COPD, incl. asthma (%) | 4.1 | 4 | 6.8 | |
| Osteoarthritis (%) | 28.0 | 28.6 | 10.8 | |
| Back problems (%) | 20.4 | 20.6 | 15.7 | < .001 |
| **Comorbidity** | | | | |
| n comorbidities Elixhauser (not mental) | 1.2 (1.4) | 1.2 (1.4) | 2.1 (1.8) | < .001 |
| Any mental comorb. (%) | 6.8 | 6.7 | 11.3 | < .001 |
| **HOSPITAL STAY** | | | | |
| In hospital last 6 months (%) | 10.9 | 10.4 | 23.3 | < .001 |
| LOS, centred for main diagnosis (days): 1st quartile (%) | 29.5 | 36.1 | 27.2 | < .001 |
| 2nd quartile (%) | 25.0 | 18.8 | 16.1 | < .001 |
| 3rd quartile (%) | 23.6 | 23.6 | 23 | n.s. |
| 4th quartile (%) | 21.9 | 21.5 | 33.6 | < .001 |

[1] Missing N = 766 [2] Missing N = 4. [3] For distribution of birth cohorts see table in S15 Table.

**Table 3. Distribution of demographic and social variables as well as readmission rates, reasons, places and duration after discharge by main diagnosis of index hospitalization.**

| Index admission | Patients (N = 62,109) | | | | | | | | | | | Readmissions (N = 2,119) | | | | | | |
|---|---|---|---|---|---|---|---|---|---|---|---|---|---|---|---|---|---|---|
| | Education (%) | | | Insurance type (%) | | Household (%) | | Gender (%) | | Age (x̄, SD) | Readmission rates (%) | CCS level 1[1] | | CCS level 2 | | Hospital | | Days until readmission (x̄, SD) |
| | low | middle | high | mandatory | (semi) private | with others | living alone | men | women | | | same | other | same | other | same | other | |
| **Chronic Condition** | | | | | | | | | | | | | | | | | | |
| Lung cancer | 34.8 | 48.4 | 16.7 | 72.1 | 27.9 | 71.2 | 28.8 | 59.6 | 40.4 | 66.9 (10.3) | 14.1 | 46.5 | 53.5 | 54.5 | 45.5 | 88.5 | 11.5 | 11.9 (8.2) |
| Colon cancer | 27.7 | 50.5 | 21.8 | 65.6 | 34.4 | 75.5 | 24.5 | 57.5 | 42.5 | 68.6 (11.8) | 5.0 | 35.7 | 64.3 | 35.7 | 64.3 | 92.9 | 7.1 | 12.4 (8.9) |
| Breast cancer | 27.5 | 52.6 | 19.9 | 60.9 | 39.1 | 71.3 | 28.7 | | 100 | 61.3 (13.0) | 2.2 | 34.7 | 65.3 | 42.6 | 57.4 | 78.2 | 21.8 | 13.9 (9.2) |
| Prostate cancer | 17.5 | 51 | 31.5 | 61.4 | 38.6 | 83.5 | 16.5 | 100 | | 68.8 (9.0) | 5.2 | 48.3 | 51.7 | 51.7 | 48.3 | 88.3 | 11.7 | 9.8 (7.4) |
| Diabetes | 38.3 | 44.5 | 17.2 | 83.4 | 16.6 | 65 | 35 | 66.7 | 33.3 | 65.4 (14.7) | 5.0 | 13.9 | 86.1 | 20.3 | 79.7 | 83.5 | 16.5 | 14.3 (9.2) |
| Acute myocardial infarction | 24.5 | 51.3 | 24.2 | 74.8 | 25.2 | 76.3 | 23.7 | 76.9 | 23.1 | 65.2 (13.0) | 3.9 | 6 | 94 | 34.9 | 65.1 | 72.3 | 27.7 | 14.7 (8.8) |
| Acute cerebrovascular diseases | 29.5 | 48.4 | 22.2 | 71.6 | 28.4 | 71.1 | 28.9 | 61.4 | 38.6 | 69.8 (13.6) | 4.6 | 18.2 | 81.8 | 39.2 | 60.8 | 80.4 | 19.6 | 14.9 (9.2) |
| Ischaemic heart disease | 25.5 | 49.6 | 24.9 | 61.3 | 38.7 | 77.6 | 22.4 | 72.8 | 27.2 | 68.6 (10.5) | 2.7 | 13 | 87 | 38.9 | 61.1 | 54.6 | 45.5 | 13.7 (9.0) |
| Congestive heart failure | 43.3 | 42.5 | 14.1 | 75.9 | 24.1 | 62.2 | 37.8 | 56.4 | 43.6 | 78.0 (10.9) | 9.3 | 25.3 | 74.7 | 38 | 62 | 85.2 | 14.8 | 16.2 (9.0) |
| COPD, incl. asthma | 40.7 | 47.3 | 12 | 77.9 | 22.1 | 61.2 | 38.8 | 51 | 49 | 68.4 (13.4) | 5.7 | 20 | 80 | 29.7 | 70.3 | 82.1 | 17.9 | 14.6 (8.5) |
| Osteoarthritis | 25.4 | 53.4 | 21.2 | 63.7 | 36.3 | 78.4 | 21.6 | 48.4 | 51.6 | 64.4 (11.3) | 1.3 | 3.1 | 96.9 | 10.5 | 89.5 | 62 | 38 | 15.0 (9.1) |
| Back problems | 26.7 | 51.7 | 21.6 | 69.4 | 30.6 | 76.3 | 23.7 | 50.3 | 49.7 | 59.6 (15.4) | 2.6 | 28.6 | 71.4 | 33.4 | 66.6 | 60.8 | 39.2 | 14.2 (8.6) |
| Total | 27.8 | 50.8 | 21.4 | 67.4 | 32.6 | 74.9 | 25.1 | 54.4 | 45.6 | 65.3 (13.4) | 3.4 | 23.9 | 76.1 | 35.6 | 64.4 | 75 | 25 | 14.1 (8.9) |

[1] Clinical Classification Software developed by the Healthcare Cost and Utilisation Project (HCUP), financed by the US Agency for Healthcare Research and Quality, adapted for Switzerland by Daniel Zahnd, University of Applied Sciences Bern.

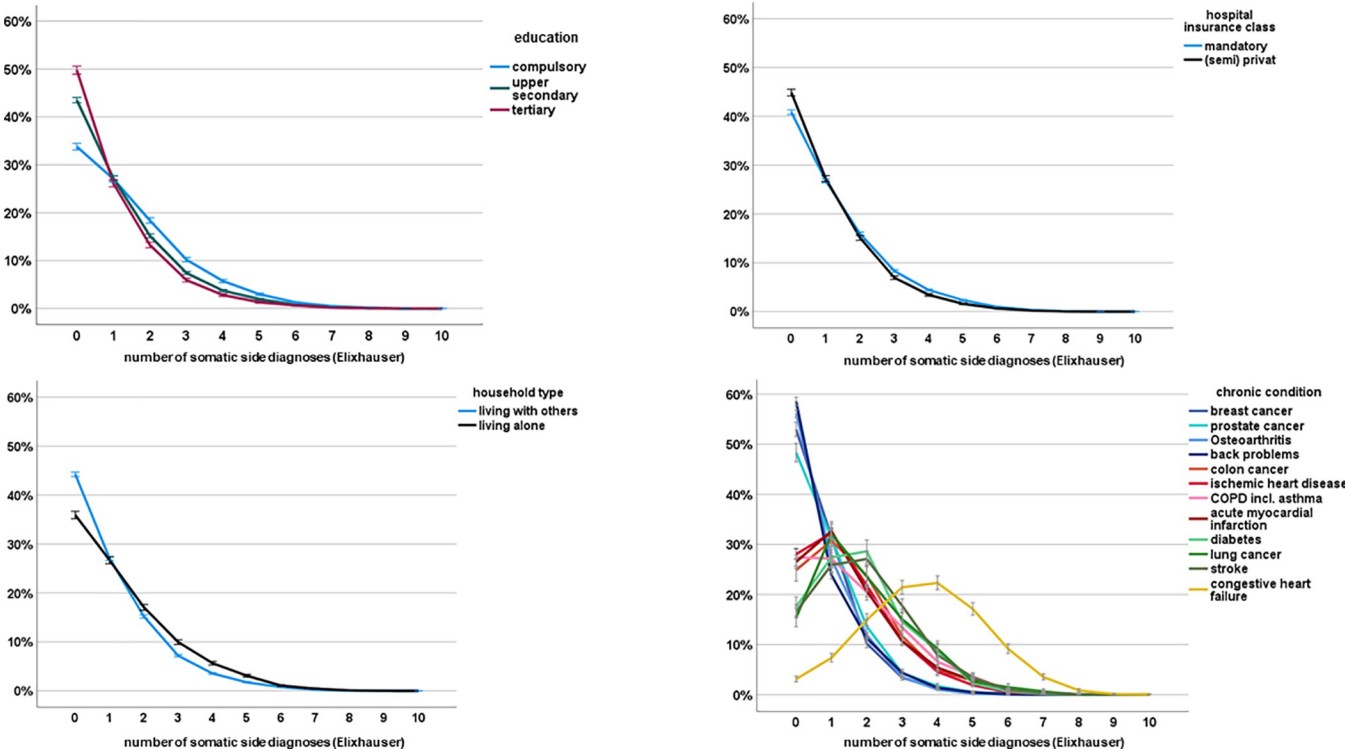

**Fig 3.** Top. Distribution (%) of number of somatic side diagnoses (Elixhauser) by educational attainment (left) and household type (right). Bottom. Distribution (%) of number of somatic side diagnoses (Elixhauser) by hospital insurance type (left) and chronic condition (right).

left), the proportion of patients with no side diagnosis was higher for patients with high vs. low educational attainment, while the proportions of those with two to five side diagnoses were clearly smaller. The same pattern was seen for the other social indicators, although the difference was less pronounced for hospital insurance type. Furthermore, Fig 3 (bottom left) shows how Elixhauser side diagnoses were distributed by chronic condition. While the majority of patients with breast/prostate cancer, osteoarthritis and back problems had none or one side diagnosis (blue lines), most other chronic conditions were accompanied by one to three diagnoses (red and green lines). Congestive heart failure showed a distinct pattern (yellow line) with most patients having three to five Elixhauser side diagnoses.

## Results of all-conditions regression models

Results of logistic regression models (Models A-C) including all chronic conditions are presented in Table 4. Social characteristics served as main predictors and demographic/health status variables and length of hospital stay as covariates.

*Social and demographic factors.* According to Model A, comparing effects of social variables, the risk for unplanned 30-day readmission was increased by 51% (95% CI: 1.31, 1.74) for patients with compulsory *education* and by 26% (95% CI: 1.11, 1.44) for patients with upper secondary education, when compared to patients with tertiary education. After adjustment for the indicators of health status (Model B), the differences remained significant but were attenuated, especially regarding the estimated effect of compulsory education compared to tertiary education. The inclusion of length of stay (Model C) did not further change effect estimates of education. As can be seen in Model A, having *(semi-) private insurance* went along with a lower risk of 30-day readmission compared to patients with mandatory insurance (OR = .81;

**Table 4. Odds ratios of multivariate logistic regression for risk of unplanned 30-day readmission by social factors, health status and length of stay in hospital (N = 62,109).**

| Outcome: risk for 30-day readmission | A: Social factors | | | | B: Health status | | | | C: Length of stay | | | |
|---|---|---|---|---|---|---|---|---|---|---|---|---|
| | Sig. | OR | 95% CI | | Sig. | OR | 95% CI | | Sig. | OR | 95% CI | |
| | | | Lower | Upper | | | Lower | Upper | | | Lower | Upper |
| Education level | | | | | | | | | | | | |
| tertiary (ref.) | < .001 | | | | 0.001 | | | | 0.002 | | | |
| upper secondary | < .001 | 1.264 | 1.111 | 1.439 | 0.005 | 1.207 | 1.059 | 1.375 | 0.005 | 1.206 | 1.058 | 1.375 |
| Compulsory | < .001 | 1.509 | 1.307 | 1.742 | < .001 | 1.316 | 1.138 | 1.523 | < .001 | 1.30 | 1.123 | 1.504 |
| Insurance class | | | | | | | | | | | | |
| mandatory (ref.) | | | | | | | | | | | | |
| (Semi-)private | < .001 | 0.813 | 0.734 | 0.902 | 0.04 | 0.896 | 0.807 | 0.995 | 0.034 | 0.893 | 0.804 | 0.992 |
| Household type | | | | | | | | | | | | |
| Living with others (ref.) | | | | | | | | | | | | |
| Living alone | < .001 | 1.233 | 1.116 | 1.361 | 0.054 | 1.104 | 0.998 | 1.221 | 0.134 | 1.08 | 0.977 | 1.195 |
| Sex | | | | | | | | | | | | |
| Men (Ref.) | | | | | | | | | | | | |
| Women | < .001 | 0.742 | 0.674 | 0.816 | 0.065 | 0.907 | 0.818 | 1.006 | 0.037 | 0.895 | 0.807 | 0.993 |
| Language skills | | | | | | | | | | | | |
| At least regional language or English (ref.) | | | | | | | | | | | | |
| Not regional language and no English | 0.076 | 1.131 | 0.987 | 1.296 | 0.323 | 1.072 | 0.934 | 1.231 | 0.426 | 1.058 | 0.921 | 1.215 |
| Age | | | | | | | | | | | | |
| < = 55 years | < .001 | | | | < .001 | | | | < .001 | | | |
| 56–65 years | < .001 | 1.551 | 1.32 | 1.823 | < .001 | 1.351 | 1.145 | 1.595 | 0.001 | 1.323 | 1.121 | 1.562 |
| 67–74 years | < .001 | 2.138 | 1.828 | 2.501 | < .001 | 1.659 | 1.408 | 1.954 | < .001 | 1.602 | 1.359 | 1.888 |
| 75+ years | < .001 | 3.362 | 2.9 | 3.898 | < .001 | 2.109 | 1.798 | 2.475 | < .001 | 1.982 | 1.688 | 2.327 |
| Chronic health condition | | | | | | | | | | | | |
| Ischaemic heart disease (ref.) | | | | | < .001 | | | | < .001 | | | |
| Lung cancer | | | | | < .001 | 5.68 | 4.59 | 7.03 | < .001 | 5.597 | 4.508 | 6.948 |
| Colon cancer | | | | | < .001 | 1.876 | 1.412 | 2.491 | < .001 | 1.845 | 1.386 | 2.455 |
| Breast cancer | | | | | 0.755 | 1.042 | 0.804 | 1.35 | 0.928 | 1.012 | 0.778 | 1.317 |
| Prostate cancer | | | | | < .001 | 2.011 | 1.603 | 2.521 | < .001 | 1.995 | 1.583 | 2.514 |
| Diabetes | | | | | < .001 | 1.796 | 1.366 | 2.361 | < .001 | 1.8 | 1.367 | 2.371 |
| Acute myocardial infarction | | | | | < .001 | 1.554 | 1.251 | 1.93 | 0.001 | 1.469 | 1.178 | 1.833 |
| Acute cerebrovascular diseases | | | | | < .001 | 1.682 | 1.342 | 2.106 | < .001 | 1.68 | 1.338 | 2.111 |
| Congestive heart failure | | | | | < .001 | 2.725 | 2.248 | 3.302 | < .001 | 2.794 | 2.299 | 3.396 |
| COPD or asthma | | | | | < .001 | 1.862 | 1.482 | 2.34 | < .001 | 1.86 | 1.475 | 2.345 |
| Osteoarthritis | | | | | < .001 | 0.56 | 0.458 | 0.684 | < .001 | 0.533 | 0.435 | 0.652 |
| Back problems and disc order | | | | | 0.16 | 1.144 | 0.948 | 1.379 | 0.264 | 1.116 | 0.921 | 1.352 |
| Comorbidity | | | | | | | | | | | | |
| NSD, centred by CHC | | | | | < .001 | 1.177 | 1.14 | 1.216 | < .001 | 1.145 | 1.108 | 1.183 |
| Mental comorbidity: no (ref.) | | | | | | | | | | | | |
| Mental comorbidity: yes | | | | | 0.002 | 1.255 | 1.087 | 1.45 | 0.019 | 1.188 | 1.028 | 1.373 |
| Previous hospital stay last 6 months | | | | | | | | | | | | |
| No (ref.) | | | | | | | | | | | | |
| Yes | | | | | < .001 | 1.777 | 1.593 | 1.983 | < .001 | 1.774 | 1.59 | 1.979 |
| Treatment in hospital | | | | | | | | | | | | |
| LOS, centred by CHC, Q1 (ref.) | | | | | | | | | < .001 | | | |
| LOS, centred by CHC, Q2 | | | | | | | | | 0.415 | 0.944 | 0.821 | 1.085 |

(*Continued*)

**Table 4.** (Continued)

| Outcome: risk for 30-day readmission | A: Social factors | | | | B: Health status | | | | C: Length of stay | | | |
|---|---|---|---|---|---|---|---|---|---|---|---|---|
| | Sig. | OR | 95% CI | | Sig. | OR | 95% CI | | Sig. | OR | 95% CI | |
| | | | Lower | Upper | | | Lower | Upper | | | Lower | Upper |
| LOS, centred by CHC, Q3 | | | | | | | | | 0.267 | 1.078 | 0.944 | 1.23 |
| LOS, centred by CHC, Q4 | | | | | | | | | < .001 | 1.637 | 1.445 | 1.855 |
| Constant | < .001 | 0.015 | | | < .001 | 0.013 | | | < .001 | 0.012 | | |
| -2 log-likelihood | 17754.37 | | | | 16825.53 | | | | 16731.89 | | | |
| ROC | 0.645 | | | | 0.73 | | | | 0.74 | | | |

95% CI: 0.73, 0.90). The association was still significant, but weaker when entering covariates (Model B and C). Regarding *type of household*, patients living alone had higher odds for 30-day readmission compared to patients living with others when controlling for other social variables (Model A). This estimated risk was more than halved when health status variables were entered (Model B). When length of stay was entered (Model C), the difference collapsed and was no longer significant. *Language skills* showed no persistent effect on the 30-day readmission rate, whereas age of patients increased the odds clearly (OR = 1.3 to 2.0 with increasing age, see Model C) and female sex reduced the odds slightly (OR = 0.90; 95% CI: 0.81, 0.99).

*Health status*. Results of Model C show, that the estimated risk for unplanned 30-day readmission was highest for patients with lung cancer (OR = 5.6; 95% CI: 4.5, 6.9) and congestive heart disease (OR = 2.8; 95% CI: 2.3, 3.4), followed by prostate and colon cancer, diabetes and COPD when compared to ischaemic heart disease, while patients with osteoarthritis showed the lowest 30-day readmission risk. On average, an increase in the number of somatic side diagnoses by one increased the risk for 30-day readmission (OR = 1.15; 95% CI: 1.11, 1.18), and having a mental comorbidity increased the risk by 19% (95% CI: 1.03, 1.37). A hospital stay in the previous six months, representing an unstable health situation, showed a rather strong effect (OR = 1.77; 95% CI: 1.59, 1.98) and the same was true for particularly long hospital stays (4th quantile) (OR = 1.64; 95% CI: 1.45, 1.86).

*Interaction analysis* pointed to differential risks for unplanned 30-day readmission by education level and household type across chronic health conditions (S1 and S2 Tables), but differences in -2 log-likelihood (32.3; 8.6) and AUC (0.003; 001) were small and model fit was not satisfactory (Hosmer-Lemeshow Test: p = 0.006; p = 0.014). We therefore ran stratified logistic regression models to identify possible condition-specific effects of social factors on 30-day readmission risks. Models were kept parsimonious, i.e., non-significant predictors (regional language skills) were excluded and conditions with small readmission numbers (colon cancer, diabetes) were not analysed separately.

## Results of condition-specific regression models

*Education level*. As shown in Fig 4 (also S3–S14 Tables), effects of social factors differed by chronic health condition. After adjustment for demographics, health status and length of stay, the odds of unplanned 30-day readmissions were increased significantly for patients with compulsory education for lung cancer (OR = 2.42; 95% CI: 1.39, 4.22), congestive heart failure (OR = 1.63; 95% CI: 1.08, 2.47) and back problems (OR = 1.53; 95% CI: 1.04, 2.25). Similar results, although statistically not significant, were observed for breast cancer, cerebrovascular diseases and COPD. For most other chronic conditions, the risk among patients with compulsory education was only slightly higher than among those with tertiary education, and it was even lower among patients with osteoarthritis. Adjusting for covariates decreased coefficients

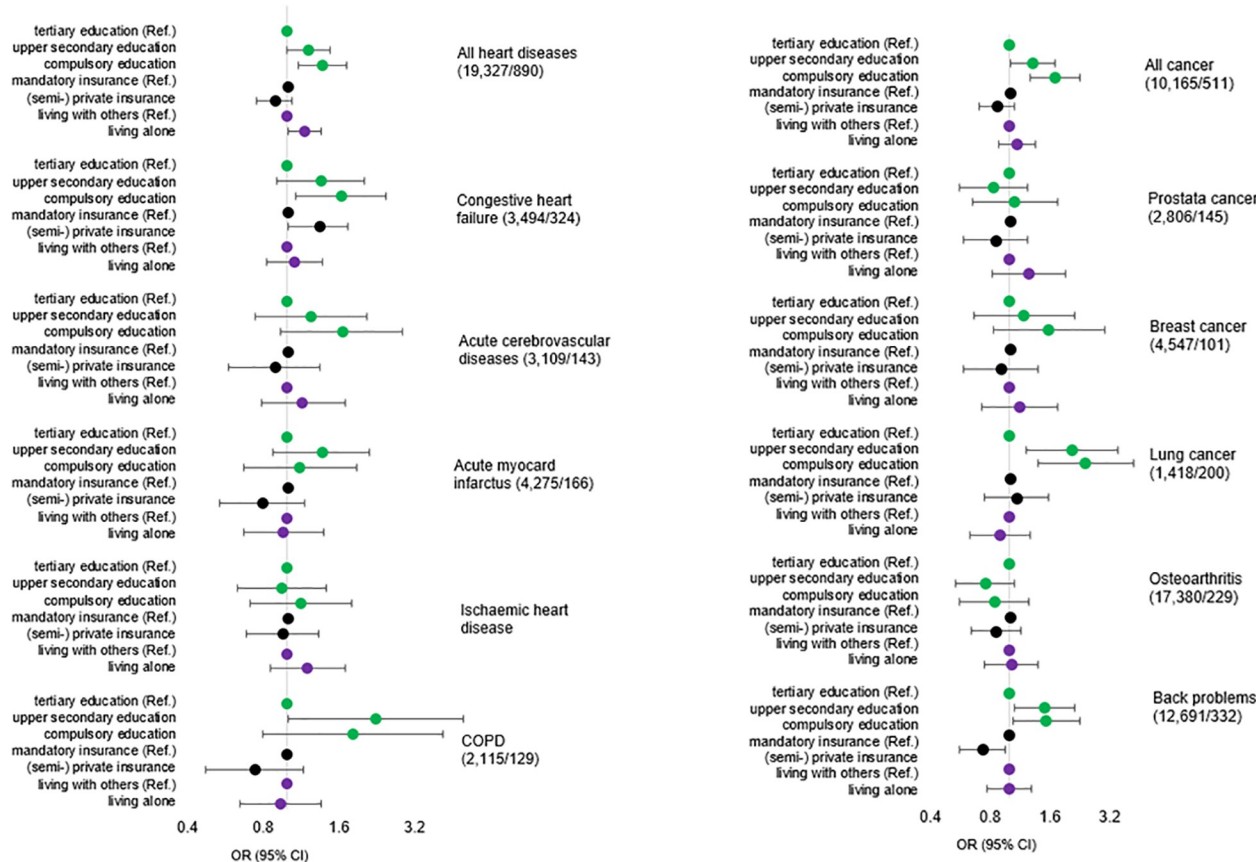

**Fig 4. Odds ratios of unplanned 30-day readmission for social factors, by chronic condition, adjusted by demographics, comorbidity, hospital admission in previous six month and length of hospital stay.**

of social factors only slightly (Model A vs. C, S3–S14 Tables). Pronounced social gradients were observed when all heart or cancer conditions were combined. The odds increased by 21% (heart: 95% CI: 0.99, 1.49) and 31% (cancer: 95% CI: 1.02, 1.69) for secondary education and by 37% (heart: 95% CI: 1.10, 1.72) and 70% (cancer: 95% CI: 1.28, 2.26) for compulsory education compared to tertiary education.

*Insurance type.* According to Fig 4, patients with (semi-) private insurance were generally at lower risk of unplanned 30-day readmission than patients with mandatory insurance, but the difference reached statistical significance only in the model for patients with back problems (OR = 0.73; 95% CI: 0.56, 0.96). Interestingly, for patients with congestive heart failure, the effect was inverted, showing an increased risk for patients with (semi-) private insurance compared to patients with mandatory insurance coverage (OR = 1.33; 95% CI: 1.01, 1.74).

*Living alone.* Patients living alone had generally higher odds for unplanned 30-day readmission than patients living together with others, but only the difference among patients with any heart conditions was significant (OR = 1.17; 95% CI: 1.01, 1.36), e.g., heart patients living alone were more at risk for 30-day readmission than heart patients living with others.

*Health status.* According to model B (S3–S14 Tables), number of Elixhauser side diagnoses showed significant effects in all types of chronic health conditions, but differences collapsed for ischaemic heart disease, myocardial infarction, COPD and osteoarthritis when adjusting for length of stay (Model C). The strongest effects were seen for cancer and cerebrovascular disease patients: Having three or more comorbidities as compared to none more than doubled

the odds for 30-day readmission for lung cancer patients (OR: 2.66, CI: 1.49, 4.75), breast cancer patients (2.42, CI: 1.17, 5.01), prostate cancer patients (2.21, CI: 1.23, 3.97) and cerebrovascular disease patients (2.48, CI: 1.27, 4.85). The presence of a mental comorbidity (Model B) increased the risk significantly in patients with ischaemic heart problems (OR: 1.99, CI: 1.10, 3.62), cerebrovascular diseases (1.67, CI: 1.08, 2.60) and back problems (1.54, CI: 1.12, 2.13) as well as in patients with any heart conditions (1.26, CI: 1.02, 1.56) or with any cancer conditions (1.58, CI: 1.14, 2.19). Differences collapsed for ischaemic heart disease and any heart disease when length of stay was added. Finally, in Model B an unstable health condition, indicated by any hospital stay in the previous six months, increased the odds for unplanned 30-day readmission more than threefold for cerebrovascular diseases (OR: 3.68, CI: 2.48, 5.45) and ischaemic heart disease (OR: 3.15, CI: 2.29, 4.35), doubled the odds for osteoarthritis (OR: 2.45, CI: 1.71, 3.50), back problems (OR: 1.99, CI: 1.50, 2.65) and any heart disease (OR: 2.16, CI: 1.85, 2.52), and increased the risk by 65% (OR: 1.65, CI: 1.02, 2.69) for myocardial infarction and 47% (OR: 1.47, CI: 1.17, 1.89) for any cancer condition.

*Length of stay*. Particularly long lengths of stay at index hospitalization were associated with higher readmission risks in all chronic conditions except for cerebrovascular diseases and COPD.

*Demographics*. In the fully adjusted models, higher age was associated with higher odds for urgent 30-day readmission in all chronic conditions but lung, breast, prostate cancer and COPD. Women had significantly lower readmission risks for osteoarthritis (OR: 0.69, CI: 0.52, 0.92) and for all cancers (OR: 0.47, CI: 0.38, 0.57) (S3–S14 Tables).

## Discussion

The aim of the present study was to analyse social inequalities in unplanned 30-day readmission rates of patients with chronic health conditions in a representative sample of patients hospitalized for acute care in Switzerland (N = 62,109). Our results point to a significant association between social factors and readmission rates for patients with chronic conditions. Risks for unplanned readmissions were significantly higher for patients with a low level of education compared to a high level (+30%) and for patients with mandatory compared to (semi-) private hospital insurance (+19%). These effects persisted when health status and length of stay were taken into account. This indicates significant differences by socioeconomic status in the likelihood of readmission after hospitalisation among patients with chronic conditions. The risk of unplanned readmission for patients with chronic conditions was, however, most strongly predicted by health status, namely previous hospitalizations before the index hospitalization (+77%) and number of comorbidities (+15% higher probability per unit increase) as well as by particularly long hospitalizations (+64%). Risks varied also strongly across chronic conditions and were particularly high among patients with lung cancer and congestive heart failure. Stratified analysis *by type of chronic condition* revealed varying importance of social factors with regard to readmission risks. Patients with low compared to high educational attainment had higher readmission risks if they had lung cancer (+142%), congestive heart failure (+63%) or back problems (+53%). Similar but not significant pattern were seen for cerebrovascular diseases, COPD and breast cancer, but not for ischaemic heart disease, myocardial infarction, prostate cancer and osteoarthritis. Having (semi-) private hospital insurance was significantly associated with unplanned 30-day readmission risks only for back problems (-27%) and CHF (+33%). We did not find evidence for an overall effect of immediate *social resources*, measured by living with others in a household, on readmission risk. Living alone was a significant predictor of 30-day readmission only among patients with any heart disease (+17%).

We believe that the effect of low socioeconomic status or, in particular, low educational attainment on the risk of unplanned 30-day readmission can be explained by factors related to social situation, such as low health literacy [15,16], inadequate coping skills [9] or stressful living conditions [13,14]. These may lead to patients having more difficulties understanding and adhering to recommended medications and therapies, communicating adequately with care providers [10] and engaging actively and responsibly in decision-making and the medical process [63]. Our analyses further show a high comorbidity burden of socially disadvantaged patients at hospital admission (Fig 3). This confirms previous study findings that, in many countries, people with low educational attainment have higher comorbidity at hospital admission [64], higher proportions of ACS conditions [65] and poorer overall health [36] than people with high educational attainment. We assume that the presence of comorbidities further complicates discharge coordination, outpatient management and also self-management of the disease, thereby exacerbating social inequalities in readmission risks.

We had assumed that social factors were particularly important for chronic diseases referred to as ambulatory care sensitive (ACS), e.g. conditions for which hospital admissions are generally assumed preventable through effective use of ambulatory care [5] and self-management, and where low socioeconomic status had been identified as a risk factor for hospitalizations [7,39,66]. However, although higher odds were observed for some chronic ACS conditions, namely congestive heart failure, no clear pattern emerged for ACS vs. non-ACS conditions. Higher readmission risks for low educated patients were observed for lung cancer and back pain, but not for common chronic ACS conditions such as angina pectoris, one of the main diagnoses of ischaemic heart disease, and COPD [5]. Although this may be partly due to small readmission numbers for some chronic conditions making it difficult to achieve statistical significance (e.g., COPD), we also assume stronger social disparities of unplanned readmission risks in severe forms or late stages of chronic conditions. Ischaemic heart disease and myocardial infarction may refer to an earlier phase of heart disease compared to congestive heart failure and cerebrovascular diseases. In the case of lung cancer, the strong educational effect might also depict the more advanced course of the illness which requires specific types of operation associated with increased readmission risks [67,68]. This assumption is supported by the fact that more than half of the unplanned 30-day readmissions of patients with lung cancer were due to health conditions related to index hospitalization whereas for other chronic conditions this proportion was much lower. Organizing nursing care after discharge from hospital to the person's home in case of severely impaired functioning might be greatly challenging for patients with a low educational attainment in Switzerland where care services are not covered by medical insurance [19]. Regarding social differences in readmission rates for patients with back problems, poor working conditions may also have contributed since these patients were more often of working age than other patients in the sample. Occupational factors, such as manual working, working overtime, and lack of supporting staff, are known to contribute to the development and the chronification of back problems [50,69]. Patients with low socioeconomic status may have fewer options to adapt work conditions or quit work to prevent worsening of back problems.

Contrary to our hypothesis, there was almost no association between living situation and risk of 30-day readmission, as we had assumed based on previous study results [28] indicating a lowering effect of social support from household members on readmission risks. The results of a qualitative study illustrate how challenging the self-management of heart failure is after discharge from hospital and what feelings this can trigger in close caregivers influencing the type of support provided. Patients reported intense emotional reactions to the onset of shortness of breath, including fear, stress, anxiety and depression. While some patients consulted a health care professional following the onset of symptoms, others reported symptoms being so

distressing that a return to hospital was seen as the safest decision. Their caregivers supported the decision to return to hospital because of the fear of negative outcomes [28]. However, as the comparison of the statistical models shows, the significant effect of living alone (Table 4, model B) only disappeared when length of hospital stay was also taken into account (model C). In our previous analysis on length of hospital stay with the same data, living alone predicted a significantly longer length of hospital stay, independent of health status, social or demographic factors [64]. We therefore assume that the effect of living alone on readmission rates is partially mediated by length of stay. In our view, this suggests that hospital staff are (can be) responsive to the needs of patients living alone, e.g., if they need extra time to organise help at home. A longer length of hospital stay also offers more time for instructions or training which may lead to a better prevention of 30-day readmission. We did not see the same pattern in patients with low levels of education; educational attainment did not predict length of hospital stay in the fully adjusted model [64].

Finally, from a methodological point of view, it should be noted that educational attainment and hospital insurance class [56] both served as measures of socioeconomic status in the SIHOS study, but educational differences seemed to capture social inequality in unplanned 30-day readmissions better than differences in insurance class. The analysis stratified by chronic condition revealed that the lower 30-day readmission risk for (semi-) private insurance was present mainly among patients with back problems, while a reverse effect was observed among patients with heart failure. We assume that this is related to the fact that a change in the law in 1996 led to a significantly lower rate of (semi-) private insurance among younger people as compared to older people. The proportion of (semi-) privately insured people in the total population almost halved from 52.5% in 1992 to 28.8% in 2017 [56]. It seems that (semi-) private insurance is a less accurate indicator of financial resources for older people than for younger people. This might bias the effect of insurance class on the readmission rate since CHF patients had the highest average age (78.0 years), while the average age of patients with back problems was lowest (59.6 years) among all chronic conditions examined in this study (Table 3). Insurance type might therefore not be a good proxy for material deprivation and low socioeconomic status over the entire age range.

## Strengths and limitations

The main strength of the present study concerns the linkage of individual-level information on demographic and social parameters to a large and representative sample of inpatients hospitalized for acute care for highly relevant chronic conditions. Unlike some of the previous studies investigating the impact of social determinants on unplanned 30-day readmission risks, our analysis was not based on aggregated data of social groups [36], restricted to one hospital [42], to a single chronic condition [31] or to readmissions to the same hospital [1]. Moreover, the use of routinely collected data kept the risk of missing data small.

In our study we found an overall rate of 3.4% for unplanned 30-day readmissions to any hospital, which is, as expected, lower than the global readmission rate of 11.1% in acute care previously estimated for Switzerland [47] and close to estimated rates of preventable (emergency) readmission rates of 3.6%-4.5% [3,47,70,71]. The lower readmission rate in this study is likely explained by the selection process (no repeated readmissions, only discharges to the patient's home, only unplanned readmissions).

The limitations of our study are mainly related to the analysis of administrative data and therefore to secondary data analysis. First of all, a possible source of time-varying measurement bias is the time span between participation in the Structural Survey and the hospital stay, which may have reached a maximum of five years. While educational attainment is expected

to be stable over time [72], type of household and language skills may have changed for some patients between participation in the SE and hospitalization. Such misclassification would likely result in bias towards the null [73] and to an underestimation of effect size. Second, we must expect unmeasured confounding. We were unable to include information on individual abilities and health behaviours or further contextual factors. Thirdly, we expect some participation bias for patients of some non-European groups due to disproportionately greater failure of data linkage, but these groups make up only a very small proportion of all patients with migration background. Fourthly, the validity of some social predictors may be limited. For instance, living alone can only be a rough indicator for social resources as it misses out on potentially supportive relationships from the wider social network of the patient. Information available on health status of patients (number of comorbidities, previous hospital stays and length of hospital stay) may also not have adequately controlled for severity of chronic condition.

## Conclusions

We conclude that the prevention of unplanned short-term hospital readmissions for chronic diseases must start from a bio-psycho-social understanding of health that takes into account the entire social situation of chronically ill patients, especially those with lung cancer, congestive heart failure and back problems, as individuals and their behaviours can only be understood in the context of their multi-layered environment. The implementation of such an understanding of health in hospital care would therefore require a systematic survey of patients' social needs with social diagnostic instruments. In order to prevent unplanned readmissions among socially disadvantaged patients, the following seems particularly necessary to us:

- As the level of education and thus health literacy emerged as a key issue in our findings, targeted measures should be taken to identify and support educationally disadvantaged patients. This would primarily include implementing early screening and documentation of health literacy at all stages of hospital care, providing discharge and self-care instructions for patients with appropriate literacy levels and empowering patients to actively engage in medical process.

- It would also include linking patients with community-based resources such as medical/ social care, educational programs and financial aid, as well as assessing patient's mental and somatic comorbidities and ensuring appropriate referral to specialised care [74]. We believe that a key element in this is an integrated team approach [22] which consists of well-functioning collaboration between health and social care professionals inside and outside the hospital. This would also provide the opportunity to develop treatment concepts based on a common understanding of how to reach and treat socially disadvantaged groups from a medical and social perspective.

- Our findings indicate once again a strong need for increased prevention activities targeted to disadvantaged population groups not only in terms of reducing risk factors for the development of chronic diseases, but also in terms of fostering early detection and early referral to health care, of removing financial or knowledge-based barriers or of counteracting "self-exclusion"-behaviours of certain population groups [75]. Again, an interdisciplinary approach is needed, involving professionals from different fields with the common goal of asserting the interests of socially disadvantaged groups within the health system, removing barriers to health care for patients with few resources, and strengthening the capacity of the Swiss health system to adequately address the needs of socially disadvantaged patients.

## Supporting information

**S1 File.**
(PDF)

**S1 Table. Odds ratios of fully adjusted logistic regression for risk of unplanned 30-day readmission (all-conditions model C) with interaction chronic condition*education level (N = 62,109).**
(PDF)

**S2 Table. Odds ratios of fully adjusted logistic regression for risk of unplanned 30-day readmission (all-condition model C) with interaction chronic condition*household type (N = 62,109).**
(PDF)

**S3 Table. Odds ratios of multivariate logistic regression for risk of unplanned 30-day readmission by social factors, health status and length of stay in hospital for all heart diseases and diabetes (N total = 19,324°/N readmissions = 897).**
(PDF)

**S4 Table. Odds ratios of multivariate logistic regression for risk of unplanned 30-day readmission by social factors, health status and length of stay in hospital for ischaemic heart disease (N total = 6,872/N readmissions = 185).**
(PDF)

**S5 Table. Odds ratios of multivariate logistic regression for risk of unplanned 30-day readmission by social factors, health status and length of stay in hospital for congestive heart failure (N total = 3,492°/N readmissions = 324).**
(PDF)

**S6 Table. Odds ratios of multivariate logistic regression for risk of unplanned 30-day readmission by social factors, health status and length of stay in hospital for acute myocard infarctus (N total = 4,275/N readmissions = 166).**
(PDF)

**S7 Table. Odds ratios of multivariate logistic regression for risk of unplanned 30-day readmission by social factors, health status and length of stay in hospital for acute cerebrovascular disease (stroke) (N total = 3,109/N readmissions = 143).**
(PDF)

**S8 Table. Odds ratios of multivariate logistic regression for risk of unplanned 30-day readmission by social factors, health status and length of stay in hospital for all cancer (N total = 10,165/N readmissions = 516).**
(PDF)

**S9 Table. Odds ratios of multivariate logistic regression for risk of unplanned 30-day readmission by social factors, health status and length of stay in hospital for lung cancer (N total = 1'418/N readmissions = 200).**
(PDF)

**S10 Table. Odds ratios of multivariate logistic regression for risk of unplanned 30-day readmission by social factors, health status and length of stay in hospital for breast cancer (N total = 4'547/N readmissions = 101).**
(PDF)

**S11 Table. Odds ratios of multivariate logistic regression for risk of unplanned 30-day readmission by social factors, health status and length of stay in hospital for prostate cancer (N total = 2,806/N readmissions = 145).**
(PDF)

**S12 Table. Odds ratios of multivariate logistic regression for risk of unplanned 30-day readmission by social factors, health status and length of stay in hospital for COPD (N total = 2115/N readmissions = 129).**
(PDF)

**S13 Table. Odds ratios of multivariate logistic regression for risk of unplanned 30-day readmission by social factors, health status and length of stay in hospital for back problems (N total = 12,691/N readmissions = 332).**
(PDF)

**S14 Table. Odds ratios of multivariate logistic regression for risk of unplanned 30-day readmission by social factors, health status and length of stay in hospital for osteoarthritis (N total = 17,380/N readmissions = 229).**
(PDF)

**S15 Table. Year of birth of patients with/without unplanned readmission (N = 62,109).**
(PDF)

## Acknowledgments

Members of the SIHOS Team: Lucy Bayer-Oglesby[1] (PI), Nicole Bachmann[1], Andrea Zumbrunn[1], Maria Solèr[1], Marcel Widmer[2], Reto Jörg[2], Carlos Quinto[3], Christian Schindler[3], Daniel Zahnd[4],

[1]Institute for Social Work and Health, FHNW School of Social Work, Olten, Switzerland; [2]Swiss Health Observatory, Neuchâtel, Switzerland; [3]Swiss Tropical and Public Health Institute, Basel, Switzerland; [4]InfoNavigation, Bern, Switzerland

*Head of the SIHOS-Team
E-mail: lucy.bayer@fhnw.ch

## Author Contributions

**Conceptualization:** Andrea Zumbrunn, Nicole Bachmann, Lucy Bayer-Oglesby, Reto Joerg.

**Funding acquisition:** Andrea Zumbrunn, Nicole Bachmann, Lucy Bayer-Oglesby.

**Methodology:** Andrea Zumbrunn, Nicole Bachmann, Lucy Bayer-Oglesby, Reto Joerg.

**Project administration:** Lucy Bayer-Oglesby.

**Visualization:** Andrea Zumbrunn.

**Writing – original draft:** Andrea Zumbrunn.

**Writing – review & editing:** Andrea Zumbrunn, Nicole Bachmann, Lucy Bayer-Oglesby, Reto Joerg.

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
