## [Decision Letter · Decision Letter 0]

15 Jun 2022

PONE-D-22-02570Social disparities in unplanned 30-day readmission rates after hospital discharge in patients with chronic health conditions: A retrospective cohort study using patient level hospital administrative data linked to the population census in SwitzerlandPLOS ONE

Dear Dr. Zumbrunn,

Thank you for submitting your manuscript to PLOS ONE. After careful consideration, we feel that it has merit but does not fully meet PLOS ONE’s publication criteria as it currently stands. Therefore, we invite you to submit a revised version of the manuscript that addresses the points raised during the review process.

We look forward to receiving your revised manuscript.

Kind regards,

Dong Keon Yon, MD, FACAAI

Academic Editor

PLOS ONE

Journal Requirements:

3. One of the noted authors is a group or consortium SIHOS Team. In addition to naming the author group, please list the individual authors and affiliations within this group in the acknowledgments section of your manuscript. Please also indicate clearly a lead author for this group along with a contact email address.

Additional Editor Comments:

Thank you for submitting your manuscript. The reviewers and I believe it is of potential value for our readers. However, the reviewers have raised a number of very important issues, and their excellent comments will need to be adequately addressed in a revision before the acceptability of your manuscript for publication in the Journal can be determined. We cannot guarantee that your revised paper will be chosen for publication; this would be solely based on how satisfactorily you have addressed the reviewer comments.

Reviewers' comments:

Reviewer's Responses to Questions

**Comments to the Author**

1. Is the manuscript technically sound, and do the data support the conclusions?

Reviewer #1: Yes

Reviewer #2: Yes

2. Has the statistical analysis been performed appropriately and rigorously? 

Reviewer #1: Yes

Reviewer #2: Yes

3. Have the authors made all data underlying the findings in their manuscript fully available?

Reviewer #1: Yes

Reviewer #2: Yes

4. Is the manuscript presented in an intelligible fashion and written in standard English?

Reviewer #1: Yes

Reviewer #2: Yes

5. Review Comments to the Author

Reviewer #1: The study is an interesting and important contribution to better understand the existing care problem and the inequality that exists in it. In the discussion, it would be great if specific recommendations were given on how re-admission could be prevented. In the abstract, the conclusion that the social differences are mainly explained by lower health literacy or low adherence falls short and neglects the differences caused by material deprivation and higher burdens.

Reviewer #2: The author's research on social disparities by using Switzerland data is generally well written and I would like to recommend accepting it if the following things can be addressed.

#1. Social disparities may affect differently between generations. I would like to see birth year distributions to study participants' generations. Also, it would be helpful for generation definitions in Switzerland.

#2. Statistical guidelines can be cited in the methodology sections. The followings are great candidates:

2-1) https://scholar.google.com/scholar?hl=ko&as_sdt=0%2C5&q=Methods+for+testing+statistical+differences+between+groups+in+medical+research%3A+statistical+standard+and+guideline+of+Life+Cycle+Committee&btnG=

2-2) https://scholar.google.com/scholar?hl=ko&as_sdt=0%2C5&q=Regression+analysis+for+continuous+independent+variables+in+medical+research%3A+statistical+standard+and+guideline+of+Life+Cycle+Committee+&btnG=

#3. One of the main factors of the LOS and readmission is hospitals. How many hospitals are involved and If there are important differences exist between them, I would like to read the reason.

6. PLOS authors have the option to publish the peer review history of their article (what does this mean?). If published, this will include your full peer review and any attached files.

Reviewer #1: No

Reviewer #2: No

---

## [Author Response · Author response to Decision Letter 0]

3 Aug 2022

Dear editor. We thank the reviewers for their valuable comments on the manuscript and we have edited the manuscript to address their concerns. Kindest regards Andrea Zumbrunn

---

## [Decision Letter · Decision Letter 1]

8 Aug 2022

Social disparities in unplanned 30-day readmission rates after hospital discharge in patients with chronic health conditions: A retrospective cohort study using patient level hospital administrative data linked to the population census in Switzerland

PONE-D-22-02570R1

Dear Dr. Zumbrunn,

We’re pleased to inform you that your manuscript has been judged scientifically suitable for publication and will be formally accepted for publication once it meets all outstanding technical requirements.

Kind regards,

Dong Keon Yon, MD, FACAAI

Academic Editor

PLOS ONE

Additional Editor Comments (optional):

This is an excellent paper.

Reviewers' comments:

Reviewer's Responses to Questions

**Comments to the Author**

1. If the authors have adequately addressed your comments raised in a previous round of review and you feel that this manuscript is now acceptable for publication, you may indicate that here to bypass the “Comments to the Author” section, enter your conflict of interest statement in the “Confidential to Editor” section, and submit your "Accept" recommendation.

Reviewer #2: All comments have been addressed

2. Is the manuscript technically sound, and do the data support the conclusions?

Reviewer #2: (No Response)

3. Has the statistical analysis been performed appropriately and rigorously? 

Reviewer #2: (No Response)

4. Have the authors made all data underlying the findings in their manuscript fully available?

Reviewer #2: (No Response)

5. Is the manuscript presented in an intelligible fashion and written in standard English?

Reviewer #2: (No Response)

6. Review Comments to the Author

Reviewer #2: All concerns have been addressed and I recommend this article as an "accept" decision on PLOS ONE. Congratulations.

7. PLOS authors have the option to publish the peer review history of their article (what does this mean?). If published, this will include your full peer review and any attached files.

Reviewer #2: No

---

## [Editor Report · Acceptance letter]

23 Aug 2022

PONE-D-22-02570R1 

Social disparities in unplanned 30-day readmission rates after hospital discharge in patients with chronic health conditions: A retrospective cohort study using patient level hospital administrative data linked to the population census in Switzerland 

Dear Dr. Zumbrunn:

I'm pleased to inform you that your manuscript has been deemed suitable for publication in PLOS ONE. Congratulations! Your manuscript is now with our production department. 

Kind regards, 

on behalf of

Dr. Dong Keon Yon 

Academic Editor

PLOS ONE